

# A citizen science approach to evaluating US cities for biotic homogenization

Misha Leong and Michelle Trautwein

California Academy of Sciences, Institute of Biodiversity Science and Sustainability, San Francisco, CA, USA

## ABSTRACT

Cities around the world have converged on structural and environmental characteristics that exert similar eco-evolutionary pressures on local communities. However, evaluating how urban biodiversity responds to urban intensification remains poorly understood because of the challenges in capturing the diversity of a range of taxa within and across multiple cities from different types of urbanization. Here we utilize a growing resource—citizen science data. We analyzed 66,209 observations representing 5,209 species generated by the City Nature Challenge project on the iNaturalist platform, in conjunction with remote sensing (NLCD2011) environmental data, to test for urban biotic homogenization at increasing levels of urban intensity across 14 metropolitan cities in the United States. Based on community composition analyses, we found that while similarities occur to an extent, urban biodiversity is often much more a reflection of the taxa living locally in a region. At the same time, the communities found in high-intensity development were less explained by regional context than communities from other land cover types were. We also found that the most commonly observed species are often shared between cities and are non-endemic and/or have a distribution facilitated by humans. This study highlights the value of citizen science data in answering questions in urban ecology.

## INTRODUCTION

Cities around the world exist in a range of environmental contexts, yet because of the requirements and preferences of their human inhabitants, they share commonalities such as landscape fragmentation, altered water and resource availability, and high densities of fabricated structures and impervious surfaces that alter climate (*Rebele, 1994*). With this ecological homogenization (*Groffman et al., 2014*) come potential consequences on the biodiversity of the organisms that live in and around cities (*Savard, Clergeau & Mennechez, 2000*). Plants have been found to bloom earlier in city centers due to the urban heat island effect (*Mimet et al., 2009*), bird migratory patterns have shifted to take advantage of resource availability (*Tryjanowski et al., 2013*), and invasive species can be more prominent because of increased rates of species introductions (*Tsutsui et al., 2000*). While such modifications are still relatively recent on an evolutionary time scale, phenotypic changes have been observed across taxa on a global scale as eco-evolutionary

Corresponding author
Misha Leong,
mleong@calacademy.org

consequences of urbanization (*Alberti, 2015*). Understanding such changes can help us better plan for future ecological dynamics in cities, such as predicting population vulnerability to invasive species or minimizing human–wildlife conflicts, such as property damage or health hazards (e.g. disease vectors).

Common ecological metrics such as species richness and abundance have shown mixed results in urban environments. A review of 105 studies on species richness along urban to rural gradients demonstrated inconsistent patterns—while some studies found that species richness decreases with higher urban intensification, other studies found the opposite (*McKinney, 2008*). Often, this greater than expected species richness can be largely attributed to non-native species (*McKinney, 2008*), highlighting the importance of additionally considering shifts in community composition. The commonality and spread of urban specialists could contribute to urban biotic homogenization—the idea that on a global scale the biodiversity of cities converges (*McKinney, 2006*; *La Sorte, McKinney & Pyšek, 2007*; *Clavel, Julliard & Devictor, 2011*). This has been particularly observed to occur with urban plants (*Schwartz, Thorne & Viers, 2006*; *Pearse et al., 2018*), and driven concerns on the cascading impacts reductions in beta diversity could have for conservation (*Socolar et al., 2016*).

A challenging aspect to measure urban homogenization is gathering sufficient data to cover the variation in ecological communities within and between cities. Within city biodiversity levels can vary greatly by neighborhood (*Sushinsky et al., 2013*). To address this, cities have frequently been examined along rural to urban gradients, although this method has been criticized for its oversimplification of features and the vagueness of definitions that makes comparisons between cities difficult (*McDonnell & Hahs, 2008*). Broad terminology like "urban" can refer to dense downtown built-up environments, residential neighborhoods, industrial areas, or parks. Even within a single type, such as residential neighborhoods, factors such as socioeconomic demographics or landscape legacy can contribute to even more local habitat heterogeneity (*Leong, Dunn & Trautwein, 2018*).

One solution to capturing all this variation and exploring patterns of biodiversity across geographically disparate cities is to utilize data generated through public engagement. Broadly referred to as citizen science (although we emphasize that one need not be a citizen of any nationality to participate), this process involves public collaboration with professional scientists in ways that help our understanding of the natural world (*Ballard et al., 2017*). Citizen science data collection overcomes the challenges of accessing private land and can be scaled up to cover multiple cities with relative ease (*Spear, Pauly & Kaiser, 2017*). There are obvious challenges such as collection biases and identification quality that need to be accounted for (*Isaac et al., 2014*), but citizen science is a potentially valuable tool that can be used far beyond science engagement or exploring expanding species distributions.

Here we examine patterns in urban biodiversity across 14 metropolitan areas in the United States using data generated by the general public. We take a multi-scale approach to examine urban biotic homogenization both between and within cities. Specifically, we ask 1) how biodiversity is shared between cities across different regions;

and 2) whether the effect of biotic homogenization gets stronger as urbanization intensifies.

## MATERIALS AND METHODS

The City Nature Challenge is a citizen science initiative started by the California Academy of Sciences and the Los Angeles Museum of Natural History that utilizes the iNaturalist platform to encourage users to photograph urban nature during a bioblitz in late April. For the 16 cities that participated in 2017 (San Francisco, CA; Los Angeles, CA; Seattle, WA; Salt Lake City, UT; Austin, TX; Houston, TX; Dallas, TX; Duluth, MN; Minneapolis, MN; Chicago, IL; Nashville, TN; Miami, FL; Raleigh, NC; Washington, DC; New York, NY; and Boston, MS) we accessed all available City Nature Challenge data from for all years available. Next, we filtered all observations to include "Research Grade" only, which is defined by the iNaturalist platform as being verifiable with a photograph and having reached a species identification consensus by at least two users in the iNaturalist community (more details available at inaturalist.org). We further filtered these observations to only include those observations that had open and un-obscured geocoordinates (geoprivacy both by user choice and for species with a conservation status are maintained on the iNaturalist platform). Because this reduced the number of available observations, we excluded the cities of Duluth and Nashville from further analyses. The 14 included metropolitan areas (Fig. 1) cover a range of geographic and environmental diversity. There were a range of number of observations between cities, highlighting the disproportionate sampling effort, with Miami having the fewest observations at 1,011 and the San Francisco Bay Area having the most at 15,733. The average number of observations of the 14 cities was 5,077 +/− 3,817. Differences in collecting effort are addressed in our analyses by using techniques such as within city comparisons and community composition metrics.

All data and scripts used for the following analyses can be found at https://github.com/mishoptera/cnc.

### Shared biodiversity between cities

We identified which species were found in the majority of the cities to compare these widespread species with the total pool of observations. We also divided the dataset by major taxa: four plant groups (monocots, dicots, ferns, and conifers) and six animal groups (birds, insects, reptiles, amphibians, mammals, and gastropods), such as to allow for better comparisons between similar taxa. To capture observed species from groups that had insufficient observations on their own (e.g. isopods, fungi, arachnids), we also created a catch-all "other" category.

### Biotic homogenization with increasing urban intensification

After seeing how biodiversity was shared between cities, we asked whether the biotic homogenization effect was stronger with increasing urbanization intensity. Based on geographic coordinates, we linked all observations with a NLCD2011 land cover classification from the Multi-Resolution Land Characteristics Consortium (MRLC).

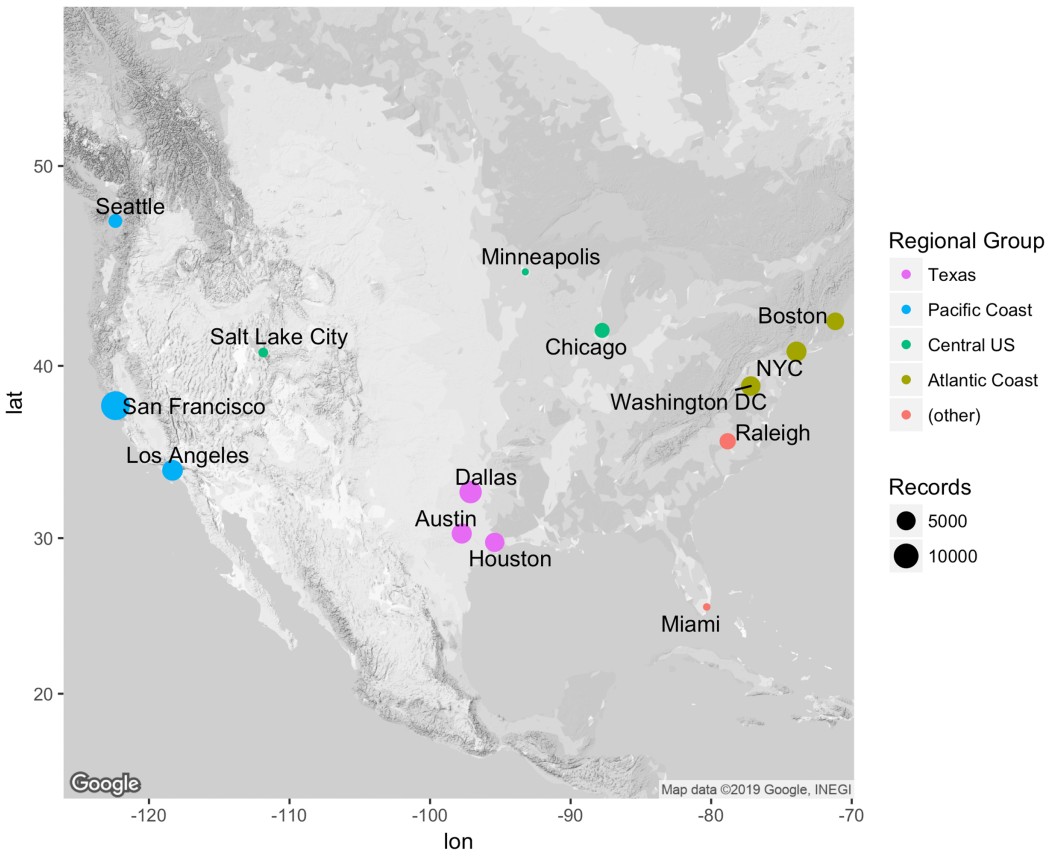

**Figure 1 Map of included City Nature Challenge cities.** The 14 cities are color grouped into major regions. The size of the circle markers represent the relative number of observations coming from each city. Miami had the fewest observations (1,011) and the San Francisco Bay Area had the most (15,733). The average number of observations of the 14 cities was 5,077 +/− 3,817.

Assessed nationwide at a 30 × 30 m resolution, every pixel is assigned one of 16 land cover classifications, four of which are forms of developed land with increasing urbanization intensity (developed-open space, developed-low intensity, developed-medium intensity, developed-high intensity; further details in Table 1). We collapsed the remaining land cover classifications into "water", "agricultural", and "natural". As we were only interested in comparing increasing levels of urbanization against the natural land use type, we excluded any observations that were classified as having occurred within agricultural or water pixels.

We then analyzed the relative influence of level of urban intensification and city on community composition. To do this, we built Bray–Curtis dissimilarity matrices based on the species composition at each level of urbanization within each city, and visualized community composition using non-metric multi-dimensional scaling (NMDS) with 100 restarts. We applied a stress cut-off of 0.20; if stress was >0.20, we considered the NMDS plot to be unreliable (*Quinn & Keough, 2002*). We visualized groupings both based on land cover type and by city.

**Table 1 Urban land cover definitions table.**

| Code | Land cover type | Description |
|------|-----------------|-------------|
| n | Natural | All areas not classified as developed, agricultural, or water. |
| d1 | Developed-open space | Areas with a mixture of some constructed materials, but mostly vegetation in the form of lawn grasses. Impervious surfaces account for less than 20% of total cover. These areas most commonly include large-lot single-family housing units, parks, golf courses, and vegetation planted in developed settings for recreation, erosion control, or esthetic purposes. |
| d2 | Developed-low intensity | Areas with a mixture of constructed materials and vegetation. Impervious surfaces account for 20–49% percent of total cover. These areas most commonly include single-family housing units. |
| d3 | Developed-medium intensity | Areas with a mixture of constructed materials and vegetation. Impervious surfaces account for 50–79% of the total cover. These areas most commonly include single-family housing units. |
| d4 | Developed-high intensity | Highly developed areas where people reside or work in high numbers. Examples include apartment complexes, row houses and commercial/industrial. Impervious surfaces account for 80–100% of the total cover. |

**Note:**
Descriptions of urbanization are based on MRLC's NLCD2011 definitions (https://www.mrlc.gov/data/legends/national-land-cover-database-2011-nlcd2011-legend).

As regional location can be an important environmental filter in determining community composition (*Williams et al., 2009*; *Aronson et al., 2014*; *Pearse et al., 2018*), we also created NMDS plots for regional groups in a series of triads of increasing geographic distance. Specifically, we focused on a Texas group (Houston, Dallas, and Austin), Atlantic Coast group (New York City, Boston, and Washington DC), Pacific Coast group (Seattle, Los Angeles, and San Francisco), and a fairly widespread Central United States group (Salt Lake City, Minneapolis, and Chicago) (Fig. 1).

To examine whether community composition becomes more similar with increased urban intensification, we subdivided observations based on their land cover classification (natural, developed-open space, developed-low intensity, developed-medium intensity, developed-high intensity). We then looked for the effect of regional location (with three city "replicates" for each region as above—Raleigh and Miami were excluded from this analysis because they did not fall neatly into one of the other regional categories). We built a PERMANOVA (Permutational Multivariate Analysis of Variance, (*Anderson, 2017*)) model for each land cover group with 999 iterations based on Bray–Curtis dissimilarity (R package *vegan* (*Oksanen et al., 2015*)), then compared the $R^2$, *p*-value and AIC score for each of the models generated by the five different land cover classifications. We would expect that if biotic homogenization were occurring with increased urban intensification, the models built off of the observations from the more developed land cover types would perform less well because the effect of regional location should be reduced.

## RESULTS AND DISCUSSION

### Shared biodiversity between cities

We analyzed 66,209 citizen science research grade iNaturalist observations across 14 US metropolitan areas. Overall, dicots, the largest plant group, were overwhelmingly the most observed (59.6%) and had the most species (52.4%). The next most observed groups were birds (12.8%), monocots (8.7%), and insects (8%). However, despite making up only 8% of the observations, insects actually made up 18.4% of the total species richness.

**Table 2 Taxa-based counts of species found in the majority of cities.**

| Taxon | Cosmopolitan pool | | Total pool | | Proportion cosmopolitan | |
|---|---|---|---|---|---|---|
| | Num species | Observations | Num species | Observations | Num species (%) | Observations (%) |
| Amphibians | 1 | 81 | 58 | 725 | 1.72 | 11.17 |
| Birds | 36 | 5,258 | 355 | 8,115 | 10.14 | 64.79 |
| Conifers | 1 | 124 | 45 | 786 | 2.22 | 15.78 |
| Dicots | 36 | 5,696 | 2,380 | 37,744 | 1.51 | 15.09 |
| Ferns | 0 | 0 | 57 | 869 | 0.00 | 0.00 |
| Gastropods | 0 | 0 | 113 | 719 | 0.00 | 0.00 |
| Insects | 7 | 1,283 | 835 | 5,067 | 0.84 | 25.32 |
| Mammals | 7 | 938 | 66 | 1,698 | 10.61 | 55.24 |
| Monocots | 1 | 33 | 499 | 5,527 | 0.20 | 0.60 |
| Reptiles | 4 | 334 | 137 | 2,123 | 2.92 | 15.73 |
| Other | 7 | 430 | 664 | 2,836 | 1.05 | 15.16 |
| Totals | 100 | 14,177 | 5,209 | 66,209 | 1.92 | 21.41 |

Birds, on the other hand, made up only 7.8% of species richness, meaning they have a higher proportion of number of observations per species.

Of the 5,209 observed species, exactly 100 were found in the majority (eight or more) of the cities (Table 2), which we hereafter refer to as our "cosmopolitan" species. While the cosmopolitan species were primarily birds and dicots (36 each), and a few mammals (seven), insects (seven), and reptiles (four), there was also one cosmopolitan species each for amphibians, monocots, and conifers, and no representative species for gastropods or ferns. Although only 1.9% of the total species richness, these widespread cosmopolitan species made up 21.4% of the total observations. Two birds, the rock dove and American crow, were the only species observed in each of the 14 cities. Ten additional species were observed in 13 cities each—seven of which were also birds (red-winged blackbird, mallard, great blue heron, turkey vulture, house sparrow, American robin, and mourning dove), but also one dicot (common dandelion), one insect (Asian lady beetle), and one mammal (common raccoon).

Taxa varied in how cosmopolitan (again, here defined as being found in the majority of our cities) they were as a group. Mammals and birds had the highest proportions of cosmopolitan species (10.6% and 10.1% respectively). On the opposite end of the spectrum, insects and dicots had a much smaller proportion of their species observed in the majority of cities (0.83% and 1.5% respectively). Our findings that cities comprise a few cosmopolitan species with a mix of many local species complement other findings that the majority of urban species are still local species (Aronson et al., 2014).

However, these cosmopolitan species accounted for the majority of observations for mammals (55.2%) and birds (64.8%), and even made up a large proportion of observations for insects (25.3%) and dicots (15.7%). While it is possible that these patterns could also be explained by cosmopolitan species being more recognizable to people (and therefore more frequently identified, leading to an inflation in the proportion of

observations for these groups), the substantial proportion of cosmopolitan species could also be indicative of a downward trend of the relative abundance of native species populations in cities. Previous multi-city studies of biotic homogenization have relied on species lists (*Aronson et al., 2014*), which cannot capture shifts in community proportions. With mass species declines in tropical and temperate ecosystems (*Hallmann et al., 2017*; *Lister & Garcia, 2018*), such findings of cosmopolitan species making up such a large portion of the community relative to native species merit further investigation.

## Biotic homogenization with increasing urban intensification

We next asked whether the effect of biotic homogenization grows stronger as a landscape becomes more developed through urbanization. The clustering in our NMDS plots suggests that urban biodiversity is to some degree city specific but also tied to particular levels of urbanization (Fig. 2). Plants exhibited a slightly different pattern from animals (Fig. S1), with the plant communities observed in the highest levels of urban intensification having the greatest differentiation, opposite to the pattern that would be expected if urban homogenization were occurring. This contrasts with a previous study that found that across cities, cultivated yards tended to be more similar to one another compared to the similarity of their associated natural areas across cities (*Pearse et al., 2018*), which could be due to being unable to differentiate between cultivated and spontaneous vegetative growth observations, and the iNaturalist platform discourages the recording of cultivated plants and animals.

We found that communities, regardless of level of urban intensification, within the same city were found close together on the NMDS plots—a pattern further reinforced by region (Fig. 2B). For example, all three Texas metropolitan cities (Houston, Dallas, and Austin) were grouped near one another, as were the cities along the Atlantic (Boston, New York City, and Washington D.C.) and Pacific Coasts (Seattle, San Francisco, and Los Angeles). Miami, being more geographically isolated and environmentally distinct than the other cities was relatively far on the plot from the other cities. Such findings complement what we found on the between cities comparison, where urban communities are largely a reflection of the local regional community, with a few cosmopolitan species. This regional clustering was found for both plants and animals. Animal communities overall were more similar between cities than plant communities, perhaps because of their mobility and ability to respond relatively quickly to land cover changes.

In the regional triad NMDS plots (Fig. 3) which peeled away some of the environmental variations between cities, community composition showed overlap between the different levels of urbanization in an ordered way along the urbanization spectrum, in that more similar levels of urbanization also share more similar communities. In all four regional groups, community composition from high-intensity urbanization were more distinct than those from all other land cover types—even more distinct than those from natural were from the least developed areas. For the Atlantic and Pacific Coast cities, there appeared to be a longitudinal gradient, with the cities falling in the geographic

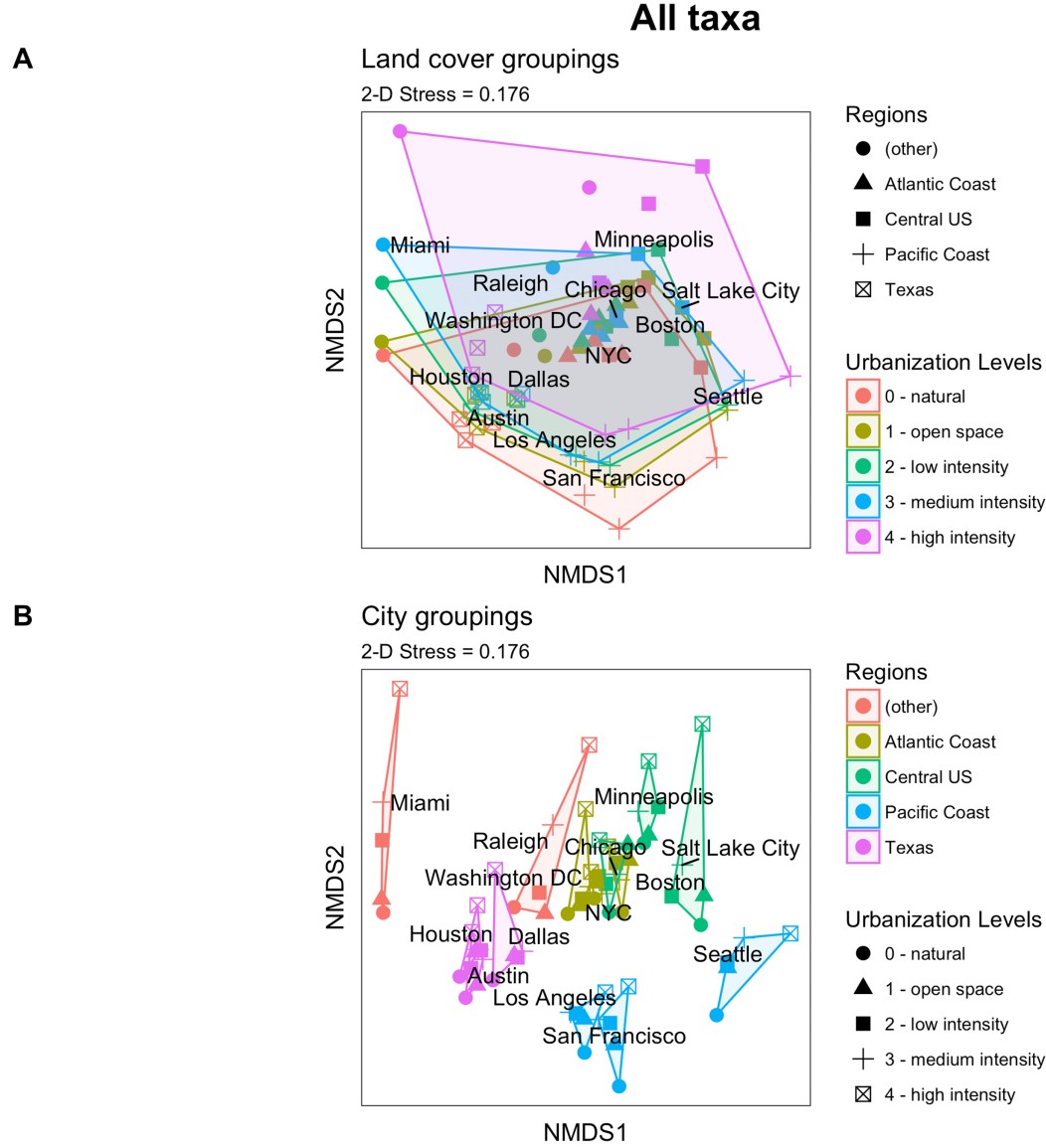

**Figure 2 Community composition NMDS plots with all taxa included.** Built from a Bray–Curtis dissimilarity matrix, each point represents the community composition of a unique combination of one of the five urbanization intensity levels in one of the 14 cities. NMDS 2-D stress = 0.176. The two plots are the same except different grouping visualizations are emphasized: in (A) points are grouped together by land cover type; in (B) points are grouped together based on city.

middle (New York City and San Francisco, respectively) having all of their land cover community compositions falling between the community compositions of cities that were more north and south. The distinctness of communities from each land cover type was more evident in those triads that have cities that are geographically closer to one another. In other words, as environmental context becomes less variable, levels of urbanization become more important in defining the community composition.

As predicted, the PERMANOVA models (for all observations, plants only, and animals only) built from observations from high-intensity land cover performed the poorest
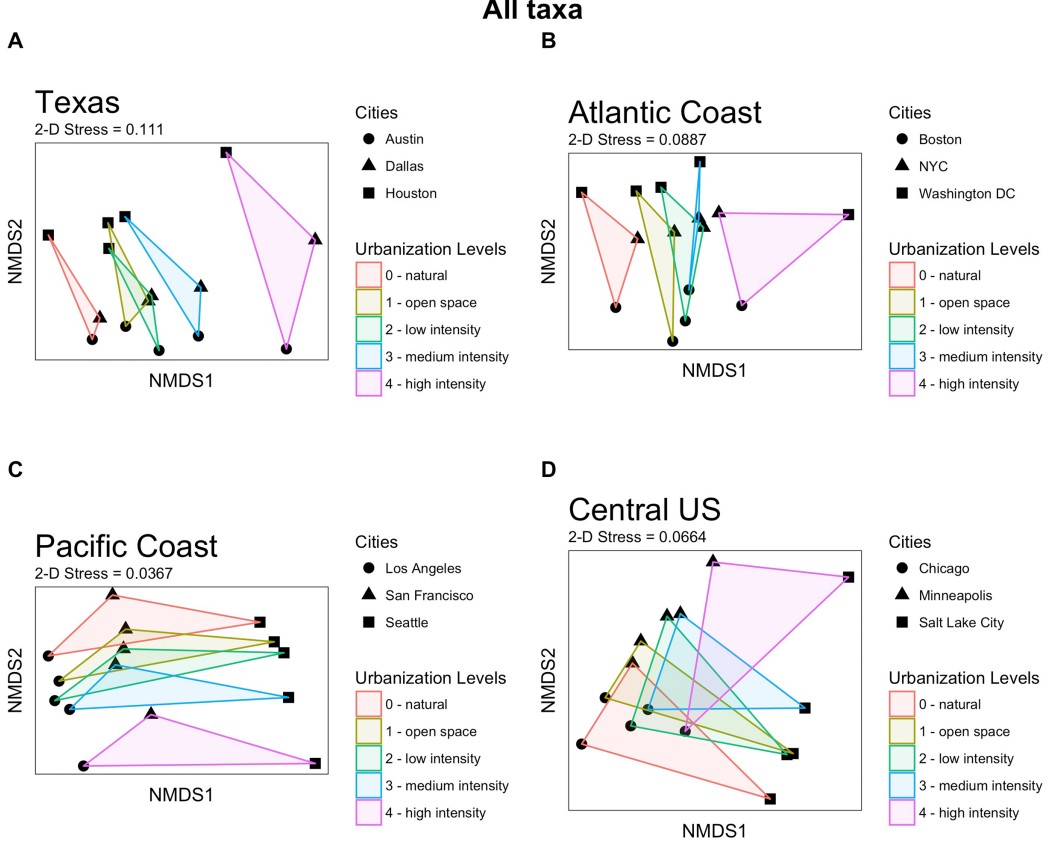

**Figure 3 Community composition NMDS plots for each regional triad with all taxa included.** Built from Bray–Curtis dissimilarity matrices, each plot represents the community composition of a unique combination of one of the five urbanization intensity levels for one of the three focal cities for each region. Plots are in order of increasing geographic distance between cities (Texas cities are ~300 km apart, whereas the Central US cities are ~1,500 km apart), and are grouped to highlight land cover type. (A) Texas (Austin, Dallas, and Houston); NMDS 2-D stress = 0.111. (B) Atlantic Coast (Boston, New York City, and Washington DC); NMDS 2-D stress = 0.0887. (C) Pacific Coast (Los Angeles, San Francisco, and Seattle); NMDS 2-D stress = 0.0367. (D) Central US (Chicago, Minneapolis, and Salt Lake City); NMDS 2-D stress = 0.0664.  

(Table 3), meaning that regional group membership was less able to predict community composition in higher intensity land cover than it could in the other land cover types. This is consistent with what we would expect to occur if biotic homogenization increases with urbanization intensification. However, the effect of regional group is still significant ($p < 0.001$) even for high-intensity land cover observations. Additionally, while the PERMANOVA models built from the high-intensity land cover observations appear the weakest (based on AIC and $R^2$), the models based on observations from natural and the other developed land cover types did not appear to decrease in strength in an ordered way with increasing urban intensification.

## Additional observations

Many species demonstrated a preferential association for either natural or high-intensity urban areas across all the cities they were found in. In general, we found that those

**Table 3 PERMANOVA results.**

| Taxon | Urban intensity | $R^2$ | *p*-value | AIC |
|---|---|---|---|---|
| All | Natural | 0.486 | 0.001 | 24.494 |
| All | Developed-open space | 0.496 | 0.001 | 23.783 |
| All | Developed-low intensity | 0.471 | 0.001 | 24.048 |
| All | Developed-medium intensity | 0.454 | 0.001 | 24.304 |
| All | Developed-high intensity | 0.400 | 0.001 | 26.237 |
| Plants | Natural | 0.488 | 0.001 | 24.977 |
| Plants | Developed-open space | 0.501 | 0.001 | 24.477 |
| Plants | Developed-low intensity | 0.472 | 0.001 | 25.168 |
| Plants | Developed-medium intensity | 0.452 | 0.001 | 25.601 |
| Plants | Developed-high intensity | 0.406 | 0.003 | 27.424 |
| Animals | Natural | 0.490 | 0.001 | 22.597 |
| Animals | Developed-open space | 0.480 | 0.001 | 22.161 |
| Animals | Developed-low intensity | 0.469 | 0.001 | 22.034 |
| Animals | Developed-medium intensity | 0.463 | 0.001 | 22.040 |
| Animals | Developed-high intensity | 0.393 | 0.010 | 24.728 |

**Note:**
We subdivided observations based on their land cover type, then looked for the effect of regional location. For each subset of observations, we built Bray–Curtis dissimilarity matrices then conducted PERMANOVA (Permutational Multivariate Analysis of Variance) analyses with 999 iterations. We repeated this for the entire dataset, plants only, and animals only.

species that favored higher intensity urban land cover tended to be non-natives, having origins in Europe, North Africa, and South Africa (e.g. common dandelion, white clover, common ivy, house sparrow, rock dove, common starling). Conversely (and expectedly), those that were found to favor more natural sites are native to North America (e.g. poison ivy, Virginia creeper, northern cardinal). However, it was difficult to identify specific ecological traits that urban specialists shared, as has been a similar finding in other urban ecology studies (*Duncan et al., 2011*).

Among the widespread cosmopolitan species we identified in the between cities comparison, we expected there to be a preferential association for the higher intensity land use types. There were in fact several species that showed this pattern—such as the house sparrow and rock dove. However, just as many widespread species favored the less disturbed natural land cover types—such as the white-tailed deer. It seems there are multiple human-associated mechanisms that act at different scales. Human transportation networks, as well as agriculture and other human-directed habitat shifts have facilitated species introductions and expanded species ranges, while urbanization has created unique habitats that allow particular species to thrive. While humans are a common denominator, species that benefit from range expansions do not necessarily also benefit from urbanization.

The western honey bee is an example of a species that varied greatly in which land cover type it favored—it was most frequently observed in the highest intensity urban land cover types in Washington DC and Los Angeles, the natural land cover types for Austin, and somewhere along the urbanization spectrum for everywhere else. The honey bee was found in every city except Minneapolis and Seattle, and was most frequently observed

in cities in Texas and California. Pollinators, and honey bees in particular, have been shown to be sensitive to climatic differences (*Gordo & Sanz, 2006*; *Bartomeus et al., 2011*), and the varying environmental conditions between cities in April could explain why the honey bee was not found in the two northernmost cities and most abundant in the more southern ones. Further, the "snapshot" approach of the City Nature Challenge captures cities at different points in their seasonal progression, as bee abundance phenology is known to vary between land cover types (*Leong et al., 2016*).

Many frequently observed species are also invasive species—such as garlic mustard. While originally introduced to North America from Europe, it thrives in the forest understory (*Stinson et al., 2006*). It was particularly abundant in Boston, New York, and Washington D.C., where it was found across all land cover types. Because there are many ongoing efforts to control this species (*Nuzzo, 1999*; *Blossey et al., 2001*), it will be important for land managers to consider that urban landscapes could also act as reservoirs maintaining sizeable populations of this species.

Our methodology utilizes within city and land cover type community composition and ranking metrics to avoid biases based on "collecting effort." However, there remain other challenges in teasing apart patterns reflecting ecological dynamics and natural history versus artifacts associated with data collected opportunistically by members of the public that currently limit ways in which we can interpret our findings. For example, species with the most observations are often not truly the most abundant species in cities, rather they are the easiest to photograph and identify (hence, the "overrepresentation" of bird taxa). Insects and other small taxa that are more difficult to photograph and identify are almost certainly under-recorded. Many species were rarely observed—2,435 of the 5,209 total species included in the dataset were singleton/doubletons, meaning they were only observed once or twice. Although we can assume that most species should be relatively equally photographable and identifiable across land cover types, we recommend using multiple approaches to make comparisons "within the biases," such as focusing on community composition and nonparametric statistical methods as we have done here.

## CONCLUSIONS

Our findings provide some support for biotic homogenization, although no single species was recorded in the highest level of urbanization across all cities. While we find that community composition is significantly impacted by degree of urban intensification, the role of geographic and environmental region seems to have a larger role in determining communities. Urban biodiversity is a mix of local natural biodiversity and introduced species that are closely associated with humans. These novel "hybrid ecosystems," with both local regional filters and the human influences of dispersal and resources are a growing reality in many parts of the world, and are continually changing with species adapting to exploit them (*Kowarik, 2011*). While it has been suggested that cities can act as reservoirs for native biodiversity (*Pearse et al., 2018*), conversely, natural areas can also be impacted by the diversity of species in the cities that they border.

Despite the complexity of urban biodiversity dynamics, this work demonstrates the power of using citizen science data in urban landscapes. The data from the City Nature

Challenge provide an opportunity to look at diverse species occurrences across many cities during the same snapshot of time in a manner that has not been possible before. The opportunistic nature of citizen science data is comparable to natural history collections in many ways (*Spear, Pauly & Kaiser, 2017*), yet with an additional factor of being focused in urban landscapes. Further, citizen science data makes up a large proportion of GBIF data and is continuing to grow at a fast rate. There are many potential future questions to explore, particularly as this dataset continues to grow in conjunction with other large environmental datasets.

While we focused our efforts using a subset of available iNaturalist observation data from the City Nature Challenge and the levels of urbanization from the National Land Cover Database, there are many more environmental and geopolitical datasets available that can be used to explore patterns in urban biodiversity. Expanding our scope to include all iNaturalist observations and museum collection specimen data could help untangle some of the complexity that we observed. Future work can also pursue broader ecological questions such as the role of climate change on urban biodiversity, phenological shifts, city connectedness, links with socioeconomics, the historical legacies of cities, and how these patterns change over time.

Finally, beyond the value that citizen science data can provide in allowing us to ask questions that would have been impossible to previously explore, the collection of these data engages the broader public in the ecological and environmental world around them in a meaningful way. An engaged network of citizen scientists is a built-in audience for science communication, making citizen science a valuable tool to increase the relevancy of environmental research. The everyday biodiversity in cities is now known to be an important contributor to city resident well-being and health. Concerns about the growing disconnect between city residents and nature can be combated (*Schuttler et al., 2018*) with increased awareness and participation in decision-making to build healthier and happier cities.

## ACKNOWLEDGEMENTS

We thank all organizers and participants of the City Nature Challenge. In particular, we are grateful to Alison Young (California Academy of Sciences), Rebecca Johnson (California Academy of Sciences), and Lila Higgins (Los Angeles Natural History Museum) as the co-founders and lead global organizers of CNC, and Amy Jaecker-Jones as the global coordinator of CNC.

### Funding

This work was supported by NSF DEB 1257960, the Doolin Foundation for Biodiversity, and the Schlinger Foundation. The funders had no role in study design, data collection and analysis, decision to publish, or preparation of the manuscript.

### Grant Disclosures

The following grant information was disclosed by the authors:
NSF DEB: 1257960.

Doolin Foundation for Biodiversity.
Schlinger Foundation.

## Competing Interests

The authors declare that they have no competing interests.

## Author Contributions

- Misha Leong conceived and designed the experiments, performed the experiments, analyzed the data, contributed reagents/materials/analysis tools, prepared figures and/or tables, authored or reviewed drafts of the paper, approved the final draft.
- Michelle Trautwein contributed reagents/materials/analysis tools, authored or reviewed drafts of the paper, approved the final draft.

## Data Availability

Data is available at GitHub: https://github.com/mishoptera/cnc.

## Supplemental Information

Supplemental information for this article can be found online at http://dx.doi.org/10.7717/peerj.6879#supplemental-information.

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
