# Peer review of "A citizen science approach to evaluating US cities for biotic homogenization"

_PeerJ, doi:10.7717/peerj.6879_

## Round 0.1 · original submission · Major Revisions

While I agree with both reviewers that the paper has merit, it is clear from the more substantive comments of Reviewer 1 that substantial revision is required before the paper can be accepted. In particular, I would ask you to consider increasing coverage of the urbanization/homogenization literature, and to address issues around the statistical analyses used in the paper.

Reviewer 1 ·

Basic reporting

Throughout I think there could be better reference to the literature. State explicitly in the introduction how this study fits into the broader field of homogenisation and urbanisation e.g. Clergeau et al. 2006 Biological Conservation, La Sorte et al. 2007 Global Change Biology, McKinney 2006 Biological Conservation (I only include these as suggestions, there is plenty of other relevant literature out there). There is also a wealth of literature on the role of traits in homogenisation in urban areas, so it would be good to at least acknowledge ecological traits within the discussion.

The results mainly rely on the interpretation of NMDS plots, of which there are a substantial number, and leads to a very qualitative description of the findings. Some of the plots could be put into supplementary material (perhaps the plant only Fig. 3) to make the paper more succinct. Clearer interpretation of the PERMANOVAs I think will improve the clarity and flow of the results.

Throughout the results, but e.g. line 164, when you refer to species diversity I assume you mean species richness. I think this term should be used throughout.

You use the term cosmopolitan (line 177) to represent species that are found in the majority of cities but are these species actually generalists? I.e. they are found everywhere. I think you could make it clear somewhere that cosmopolitan species are not necessarily exclusively found in high intensity land cover types.

I have a number of issues with Figure 5. Firstly, the legend could be clearer, adjust your ggplot code so that you just have dots, not dots overlaid on ‘a’. In 5b, I can’t distinguish Red-winged blackbird from Mallard, are they overlapping?

Overall, 5a and 5b are difficult to interpret. Were the slopes of the CAM and ARM with intensification always significant? The analyses that lead to the creation of Figure 5, as well as the interpretation of this figure could both be expanded upon to improve comprehension. What statistical tests have been done to generate this? What is it actually showing? Is there an overall relationship?

Experimental design

I’d like to know what taxa fell into the ‘other’ category. Can you give a couple of examples? (Line 114)

Add reference to justify use of 0.20 as stress cut-off in NMDS e.g. Quinn and Keogh 2002 Experimental design and data analysis for biologists. – Cambridge Univ. Press

Design is relatively clear but interpretation of how this reflects homogenisation could be stated more explicitly within methods. For example, how do city accumulation metric and averaged ranking metric reflect biotic homogenisation? To my knowledge these aren’t commonly used measures, and it took me a few reads to get my head around what they mean. Although they are detailed in the supplementary material, more detail on their interpretation is necessary within the main text.

Further, in line 142-143, it would be good to have a brief explanation/example on the assumption that urban specialists are contributing to homogenisation.

Validity of the findings

I am not convinced that you can use this data for community composition and then calculate Bray-Curtis. In line 105 it is stated that differences in collector effort are addressed in the analyses, but I don’t see how. I understand that the ARM accounts for recorder effort in the species level approach, but how is it accounted for in the community approach. If the observations are used to build the Bray-Curtis dissimilarity matrix, then are you assuming that number of observations = relative abundance? If recorder effort has been accounted for, I think this needs to be made much more explicit.

The statement in line 203-204 that city explained more than land cover type needs to be justified numerically in the text. Is this based on the R2?

Additional comments

Overall I think this is an interesting piece of work, but the presentation of the study could be improved.
Theories underlying the paper could be fitted better within the current literature. The reference list is by no means exhaustive so more detail about how the current approach compares to alternative (and more common) approaches within the literature and how their results fit into the wider discussion of urbanisation effects on community homogenisation.

Some minor comments:
Keep use of abbreviations consistent e.g. line 152
Check grammar e.g. lines 198, 238 and 265-266
Lines 232-236 This seems to be more methods than results
Use e.g. instead of ex. throughout
Splitting the results and discussion would make the paper easier to follow and better put the results in the context of the current literature. It would also fit the format of the journal. Also add conclusions section to abstract.

·

Basic reporting

No comment

Experimental design

No comment

Validity of the findings

No comment

Additional comments

A great contribution to the urban homogenization literature. Clearly citizen science and iNat can be a huge source of data. The problem is the bias factor but your discussion correctly identifies potential influences. In this case, your finding that relative abundance may play a big role (e.g., decline of natives) is really interesting and often neglected but could be strongly affected by the fact that citizens tend to photograph large, colorful or otherwise conspicuous species so that abundance is inflated in those species. Nevertheless, this study is a great first step in describing the visible patterns. Your finding that urban composition is driven by a few cosmopolitan and many local/regional species adapted to humans is compelling and critically important to our basic understanding of urbanization impacts on biodiversity. Your analysis is well conceived, implemented and easy to follow.

---

## Round 0.2 · accepted · Accept

I am satisfied that you have made substantial efforts to improve your paper in line with the comments of Reviewer 1, and I have no further editorial comments at this time.

#